# Urgent Hospitalizations Related to Viral Respiratory Disease in Children during Autumn and Winter Seasons 2022/2023

**DOI:** 10.3390/v15122425

**Published:** 2023-12-14

**Authors:** Francesca Peri, Beatrice Lorenzon, Carolina Cason, Alessandro Amaddeo, Stefania Norbedo, Manola Comar, Egidio Barbi, Giorgio Cozzi

**Affiliations:** 1Department of Medicine, Surgery and Health Sciences, University of Trieste, 34100 Trieste, Italy; beatricelorenzon@gmail.com (B.L.); egidio.barbi@burlo.trieste.it (E.B.); 2SSD of Advanced Microbiology Diagnosis and Translational Research, Institute for Maternal and Child Health-IRCCS Burlo Garofolo, 34100 Trieste, Italy; carolina.cason@burlo.trieste.it (C.C.); manola.comar@burlo.trieste.it (M.C.); 3Emergency Department, Institute for Maternal and Child Health-IRCCS Burlo Garofolo, 34100 Trieste, Italy; alessandro.amaddeo@burlo.trieste.it (A.A.); stefania.norbedo@burlo.trieste.it (S.N.); giorgio.cozzi@burlo.trieste.it (G.C.); 4Department of Medical Sciences, University of Trieste, Piazzale Europa 1, 34127 Trieste, Italy; 5Department of Pediatrics, Institute for Maternal and Child Health IRCCS “Burlo Garofolo”, 34100 Trieste, Italy

**Keywords:** Respiratory Syncytial Virus (RSV), bronchiolitis, emergency department

## Abstract

Aim: The loosening of social distancing measures over the past two years has led to a resurgence of seasonal epidemics associated with respiratory viral infections in children. We aim to describe the impact of such infections through urgent hospitalizations in a pediatric emergency department. Methods: We performed a retrospective review of medical records of all children and adolescents with a positive nasal swab admitted at the children’s hospital IRCCS Burlo Garofolo of Trieste, in Italy, from September 2021 to March 2022, and September 2022 to March 2023. Results: Respiratory Syncytial Virus and Influenza viruses accounted for up to 55% of hospitalizations for respiratory infections during the study periods. During the last season, the number of hospitalizations related to the Influenza virus was five times higher than those related to SARS-CoV-2 (25% vs. 5%). Respiratory Syncytial Virus was associated with a greater need for respiratory support, mostly HFNC (High Flow Nasal Cannula). Conclusions: Respiratory Syncytial Virus and Influenza virus had a more significant impact on urgent hospitalizations during the past wintery seasons than SARS-CoV-2.

## 1. Introduction

Since 2020, the introduction of social distancing measures limiting the spread of the COronaVIrus Disease 19 (COVID-19) pandemic has wholly changed the epidemiology of common viral infections in children. These measures included a national lockdown with school closures from March to May 2020, followed by the mandatory use of face masks and different levels of social distancing measures from November 2020 to May 2021. These measures have had a profound impact on the transmission of viruses through respiratory droplets [1]. During the first winter of the COVID-19 pandemic, the disappearance of winter epidemics sustained by Respiratory Syncytial Virus (RSV) and Influenza virus was observed. In contrast, the prevalence of Rhinoviruses remained largely unaffected, likely due to their viral stability and the role of asymptomatic transmission [2,3,4,5]. In the next two years, the progressive loosening of most social distancing measures led to a resurgence of seasonal epidemics associated with respiratory viral infections in children. Initially, several European countries reported out-of-season RSV outbreaks during the spring of 2021 [6,7]. At the same time, an RSV epidemic was observed during the autumn of 2021, showing an increase in disease severity [7,8,9]. Finally, the first Influenza virus epidemic was reported in the autumn of 2022, along with RSV and the COVID-19 pandemic [10]. The burden on urgent hospitalizations related to the unprecedented combination of three highly diffusible and virulent infections has not yet been described. This report aimed to describe the impact of different respiratory viruses on urgent hospital admissions in the north-east of Italy, with particular focus on RSV, Influenza virus and SARS-CoV-2 during the autumn–winter season in 2022/23 compared to the same period in the year before.

## 2. Methods

We conducted a retrospective study reviewing the medical records of all children and adolescents hospitalized at the pediatric emergency department (PED) at the tertiary level, in the University Teaching Children’s Hospital, Institute for Maternal and Child Health, IRCCS Burlo Garofolo of Trieste, Italy, from 1 September 2021 to 31 March 2022 and from 1 September 2022 to 31 March 2023. Since the summer of 2021, all children attending the PED with fever or respiratory symptoms and who needed hospitalization underwent a multiple nucleic acid amplification assay using a nasopharyngeal (NP) swab for 13 common viral respiratory agents, including RSV, SARS-CoV-2, Influenza virus and Rhinovirus (Respiratory Flow Chip assay—Vitro, Sevilla, Spain).

Eligible patients were children and adolescents, from 0 to 17 years of age, who were admitted and hospitalized for at least 24 h at the PED. Among eligible patients, all cases that tested positive for a viral agent at the swab test were retained for analysis. For every selected patient, the following data were recorded: age, gender, diagnosis and the need for ventilatory support. The triage code at arrival was also registered. Italian pediatric triage codes consist of a five grade scale; for the purpose of our study, “urgent code” was the highest clinical priority code. The number of patients who arrived at the PED during the two study periods and the number of viral swabs performed were also collected.

The primary study outcome was the number of patients hospitalized for RSV, Influenza virus or SARS-CoV-2 infection between the 2021/2022 and 2022/2023 autumn and winter seasons.

### 2.1. Ethics

The Institutional Review Board (IRB) of the Institute gave ethical approval to the study protocol (RC 10/2020). Due to the study’s retrospective nature, no specific written informed consent was administered.

### 2.2. Statistical Analysis

Data of enrolled children were summarized using descriptive analysis. Categorical variables were reported through absolute frequencies and percentages. 

We divided the study cohort into two periods according to the patient’s arrival date: 1 September 2021–31 March 2022 and 1 September 2022–31 March 2023.

The chi-square and Fisher’s Exact tests were used to examine the variables of interest in the differences between the two periods. Data were entered into an Excel spreadsheet, and statistical analyses were performed with R software (version 4.0.3, 2020). Statistical significance was considered for *p*-values < 0.05.

## 3. Results

During the study period, 30.718 PED visits were performed: 13.706 (44.6%) in autumn–winter 2021/22 and 17.012 (55.4%) in autumn–winter 2022/23. Of these, 97 (0.7%) and 188 patients (1.1%) with a positive multi-viral NP swab were hospitalized during autumn–winter 2021/22 and 2022/23, respectively. Forty-six patients (47%) were female in the first group, and eighty-two (44%) in the second. The mean age was significantly lower in the first period (2.0 years ± 3.3) compared to the second period (3.1 years ± 4.3) (*p* = 0.028). Table 1 summarizes the impact of RSV, Influenza virus and SARS-CoV-2 on urgent hospitalizations during the autumn–winter season 2022/23, compared to the same period the year before. During autumn and winter 2021/2022, three urgent hospitalizations related to Influenza virus infection were recorded. On the other hand, the number of urgent hospitalizations related to the Influenza virus rose to five times higher than those related to SARS-CoV-2 over the last winter season, 48 (25%) vs. 10 (5%). RSV and Influenza viruses accounted for up to 55% of hospitalizations for respiratory infections. In our population, RSV accounted for most urgent hospitalizations in the autumn–winter season 2021/2022, 27 (28%) and, 57 (30%) in the following season. Rhinovirus was the most prevalent in our population in both seasons: 38 (39%) in 2021/22 and 75 (40%) in 2022/23. It played a dual role as a cause of both coinfections and primary infections, accounting for 31 cases of isolated infection in 2021/2022 and 11 cases in 2022/2023. During autumn–winter 2021/2022, six cases (19%) were associated with bronchiolitis and four (12%) required respiratory support with HFNC. In the subsequent season, two cases (18%) resulted in bronchiolitis and only one (9%) required HFNC support. No patients in either season required support with CPAP. 

The few cases of coinfections did not allow a dedicated subgroup analysis. Nevertheless, we did not record a greater severity in those patients with coinfections. 

Table 2 summarizes clinical data according to virus group, comparing the impact of each virus on the urgent triage code and the need for respiratory support between the two years. In our cohort, RSV was responsible for most of the urgent triage codes during the autumn–winter seasons of 2021/2022 and 2022/2023: 14 (38%) and 30 (49%), respectively. However, it must be highlighted that Rhinovirus was responsible for urgent triage codes in 31 (84%) cases during the autumn–winter season 2021/2022. It represents the cause of 67% of bronchiolitis during both 2021 and 2022 and was associated with greater need for respiratory support, mostly HFNC (7, 26%, during 2021 and 21, 37%, during 2022). No statistical differences were found between the two groups. 

## 4. Discussion

Autumn and winter seasons 2022/2023 were the first seasons after the beginning of the COVID-19 pandemic during which an Influenza virus epidemic along with RSV and SARS-CoV-2 was observed in Europe [10]. This study highlighted the impact of this epidemic on urgent hospitalizations in children as related to other common viral agents. 

We observed a significant growth in hospital admissions (+25%) and a higher rate of urgent hospitalizations (+51%) during autumn–winter 2022/23 compared to the previous year, and one of the determining factors appeared to be the resurgence of Influenza. The Influenza virus was responsible for many urgent hospitalizations with similar rates due to other viral agents between 2021 and 2022. Moreover, we detected a higher prevalence of hospitalizations and a more substantial impact of RSV bronchiolitis than SARS-CoV2 during autumn–winter 2022–2023. More specifically, our data aligned with other recent studies, as RSV was responsible for a greater use of respiratory support (HNFC and CPAP) in our setting compared to other viruses [11]. Whether this could suggest a more confident use of HFNC and CPAP by pediatricians outside an intensive care setting rather than an increase in severe forms of bronchiolitis is still a matter of debate. Our data support a change in the management of RSV-bronchiolitis that seems to have gained a relatively greater gravity if compared with the others respiratory infections, emerging as the primary cause of urgent hospitalization and the need for respiratory support. As demonstrated in previous studies, respiratory supports and HFNC increased significantly compared to the pre-pandemic era, but the rate of intubation and the length of stay did not change, suggesting a more aggressive treatment attitude rather than a more severe disease [12]. 

On the other hand, no subjects with SARS-CoV-2 positivity required respiratory aid during the last season and only one patient required it during autumn–winter 2021–2022. Our findings also aligned with previous data, showing that SARS-CoV-2 infection has a respiratory involvement significantly milder in children than in adults [13,14]. Pediatric data showed that SARS-CoV-2 only rarely causes symptoms suggestive of bronchiolitis [15]. Our data help to offer further insights into the natural history of SARS-CoV-2 in children compared to other viral agents such as Influenza and RSV. In addition, this work confirmed that respiratory support is only rarely needed in children admitted with SARS-CoV-2 infection, as already reported in previous pediatric works [15] and in extraordinary contrast with adults [16].

Finally, our data, in line with previous studies, revealed a consistent prevalence of Rhinovirus in both seasons, constituting approximately half of all viral detections. While Rhinovirus frequently co-occurs with other respiratory viral infections, our findings indicate that, despite an overall rise in its prevalence during the 2022/2023 season, the number of hospitalizations solely attributable to this virus has decreased in comparison with the previous autumn–winter season. The underlying reasons for these findings remain unclear and may be attributed to a combination of virological, environmental and behavioral factors, including the stability of these non-enveloped viruses on surfaces and their prominent transmission routes [17].

The resurgence of common viral infections was somewhat predictable, especially after the relaxation of social distancing measures, mainly school closures and facial mask wearing. On the other hand, the impact on urgent hospitalizations during the resurgence of common epidemics such as Influenza virus and RSV was less predictable. Moreover, until autumn 2022 no comparisons were possible between the Influenza virus and SARS-CoV-2. 

This study has several limitations. Due to the study design, some cases may have been missed or mislabeled. Moreover, our data referred to a PED of a single Institution, so the generalizability is limited. According to the policy of our Institute, we did not perform the nasopharyngeal swab for multi-viral tests on all the patients who arrived at the PED during the study periods, but only on patients who needed urgent hospitalization. Therefore, we cannot provide the distribution of viral infections among the general population accessing the PED during the study periods. On the other hand, before the COVID-19 pandemic, the execution of viral tests in our setting was extremely rare. Therefore, we could not provide a measure of the impact of the Influenza virus and RSV on urgent hospitalizations in our setting at that time. We reported data related to the first epidemic of the Influenza virus after the pandemic, and the mean age of enrolled children was very young, so we can presume that this was the first infection with this virus for most of the children in our population, and this may have influenced the study results. No data on flu vaccination in our population were available; considering the mean age of <5 years of the children in our study, it is plausible to say that the majority of our patients were not vaccinated for SARS-CoV-2, given that vaccination was not available for this age group. Furthermore, we did not analyze the subgroup of patients with comorbidities, as these did not appear to influence the final outcome or the need for respiratory support, which was necessary in previously healthy infants with bronchiolitis.

Finally, this study was performed during the spread of the SARS-CoV-2 Delta (winter season 2021) and Omicron (winter season 2022) genetic variants throughout Italy. The cases positive for SARS-CoV-2 in this study were sustained by those variants, but no genetic data were collected. 

In conclusion, this study showed that after three years of the COVID-19 pandemic, Influenza virus and RSV infections had a more significant impact on urgent hospitalizations than SARS-CoV-2. According to other Italian centers, our data may provide support for national vaccination strategies [18].

Since the beginning of the COVID-19 pandemic, remarkable scientific and public health attention has been focused on SARS-CoV-2, sometimes at the expense of measures to counteract other common viral infections in children. 

Future studies are needed to maintain surveillance in the forthcoming winter season. 

## Figures and Tables

**Table 1 viruses-15-02425-t001:** Virological data for urgent hospitalizations.

	2021/22	2022/23
Number of PED visits	13.706	17.012
Number of urgent hospitalizations	110	226
Number of urgent hospitalizations with positive NP swab	N = 97 * RSV = 27 (28%) Influenza = 3 (3%) SARS-CoV-2 = 22 (23%)	N = 188 * RSV = 57 (30%) Influenza = 48 (25%) SARS-CoV-2 = 10 (5%)
	Others: Rhinovirus = 38 (39%) Parainfluenza virus = 3 (3%) Adenovirus = 8 (8%) B. Pertussis = 0	Others: Rhinovirus = 75 (40%) Parainfluenza virus = 7 (4%) Adenovirus = 14 (7%)
Co-infections	RSV + rhinovirus = 5 (5%) RSV + SARS-CoV-2 = 1 (1%) RSV + other coronaviruses = 2 (2%) SARS-CoV-2 + rhinovirus = 1 (1%)	RSV + rhinovirus = 15 (8%) RSV + SARS-CoV-2 = 1 (0.5%) RSV + influenza = 4 (2%) SARS-CoV-2 + influenza = 0 Influenza + rhinovirus = 11 (6%) SARS-CoV-2 + rhinovirus = 0

List of abbreviations: pediatric emergency department = PED, nasopharyngeal = NP, respiratory syncytial virus = RSV, severe acute respiratory syndrome coronavirus 2 = SARS-CoV-2. * The total number refers to the number of positive tests; the sum of individual viruses is higher due to co-infections.

**Table 2 viruses-15-02425-t002:** Demographic characteristics of the population according to virological tests.

	RSV			Influenza			SARS-CoV-2			Rhinoviruses	
	2021/2022	2022/2023	*p*	2021/2022	2022/2023	*p*	2021/2022	2022/2023	*p*	2021/2022 2022/2023	*p*
Age, years (mean, SD)	1.2 ± 2.1	1.9 ± 3.8	0.28	3.5 ± 0.8	4.9 ± 5	0.12	1.1 ± 3.7	2 ± 4.7	0.6	2.9 ± 4.1 2.2 ± 3.4	0.61
Urgent triage code	14 (38%)	30 (49%)	0.44	-	12 (20%)	N/A	3 (8%)	3 (5%)	0.18	31 (84%) 11 (18%)	0.001
HFNC	7 (26%)	21 (37%)	0.18	-	5 (10%)	N/A	1 (4%)	-	N/A	4 (12%) 1 (9%)	0.76
CPAP	5 (18%)	2 (3%)	0.98	-	0	N/A	0	-	N/A	0 0	N/A
Bronchiolitis	18 (67%)	37 (65%)	0.49	-	6 (12%)	N/A	2 (9%)	-	N/A	6 (19%) 2(18%)	0.94

Abbreviation list: HFNC = high flow nasal cannula, CPAP = continuous positive airway pressure, RSV = respiratory syncytial virus, SARS-CoV-2 = severe acute respiratory syndrome coronavirus 2.

## Data Availability

Data are contained within the article.

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
