# Peer review of "Urgent Hospitalizations Related to Viral Respiratory Disease in Children during Autumn and Winter Seasons 2022/2023"

_viruses, 2023, doi:10.3390/v15122425_

Round 1

Reviewer 1 Report (Previous Reviewer 4)

Comments and Suggestions for Authors

None.

Author Response

Thank you for your revision.

We further reviewed our manuscript.

Reviewer 2 Report (Previous Reviewer 2)

Comments and Suggestions for Authors

The manuscript is presenting the differences regarding urgent hospitalizations related to various respiratory viruses in children during two consequtive seasons. Three viruses are outlined: Influenza virus, Respiratory Syncytial Virus and SARS-CoV-2  The finding ot this study are presenting data of which are clinicians well aware, however, observed things need to be reported and supported with exact numbers. Therefore, this article is possible valuable scientific addition to the existing knowledge in this field.

My remarks:

- in Introduction, I suggest retaining ref. https://pubmed.ncbi.nlm.nih.gov/36560751/ since it is not clear weather this reference is cited or not in the final version of the manuscript

- Rhinosiruses were most prevalente in both observed periods. However, their impact was presented in the main text but not in the Table 2. I suggest to add that and to discuss their impact additionaly in Discussion.

- Please elaborate how were the percentages calculated, for example: Urgent triage code in RSV: 30 (49%) - In this period there were 57 RSV+ patients and it not clear to me since 30 out of 57 makes 52.6%

- The authors state that the reason for 116 more urgent hospitalizations in the second period is Influenza. Indeed, in that second period 45 more children than in the first period were hospitalized due to Influenza infection. However, 37 more was due to Rhinovirus without mentioning that in Discussion. 

- Also, in Discussion, additionally elaborate and add more data supporting the statement that "our data provide support for a change in clinical severity of RSV bronchiolitis..." I still not understand completely how the change of RSV severity is evaluated.

Author Response

- in Introduction, I suggest retaining ref. https://pubmed.ncbi.nlm.nih.gov/36560751/ since it is not clear weather this reference is cited or not in the final version of the manuscript

Answer: Thank you. We corrected the bibliography.

- Rhinosiruses were most prevalente in both observed periods. However, their impact was presented in the main text but not in the Table 2. I suggest to add that and to discuss their impact additionaly in Discussion.

Answer: Thank you. We added the data related to Rhinoviruses in Table 2.

- Please elaborate how were the percentages calculated, for example: Urgent triage code in RSV: 30 (49%) - In this period there were 57 RSV+ patients and it not clear to me since 30 out of 57 makes 52.6%

Answer: The percentages were calculated as follows:

% urgent triage code per virus: n. of urgent triage cose virus-related/ total of urgent triage code

% bronchiolitis: n. of bronchiolitis virus-related/ total of virus-related admissions

% CPAP and HFNC: n. CPAP/HFNC virus-related/ total of virus-related admissions

- The authors state that the reason for 116 more urgent hospitalizations in the second period is Influenza. Indeed, in that second period 45 more children than in the first period were hospitalized due to Influenza infection. However, 37 more was due to Rhinovirus without mentioning that in Discussion. 

Answer: We have explained in the Discussion how the overall prevalence of Rhinovirus has indeed increased, but due to co-infections. Meanwhile the proportion of urgent hospitalizations where it was found as the sole causative agent has decreased.

- Also, in Discussion, additionally elaborate and add more data supporting the statement that "our data provide support for a change in clinical severity of RSV bronchiolitis..." I still not understand completely how the change of RSV severity is evaluated

Answer: Thank you, we reformulated the sentence, clarifying that there has not been an actual worsening of the clinical severity of RSV bronchiolitis. Instead, there is a relative increase in severity when compared to other respiratory viruses and a more aggressive treatment then in the pre pandemic era.

Round 2

Reviewer 2 Report (Previous Reviewer 2)

Comments and Suggestions for Authors

The whole text should be checked by English native speaker before resubmitting it again.

The number of patients should be presented differently. I do not think, for example n.6, is appropriate way of presenting data.

It is stated "Table 2 summarizes clinical data according to virus group, comparing the impact of each virus on the urgent triage code and the need for respiratory support between the two years. In our cohort RSV is responsible for most of the urgent triage code during autumn-winter seasons 2021/2022 and 2022/2023, respectively n. 14 (38%) and n. 30 (49%)." However, when you analyze Table 2 it is clearly visible that in 31 (84%) cases during 2021/22 Rhinovirus was responsabile for urgent triage code.

I do not understan clearly difference between urgent hospitalization and urgent triage codes? It is stated "the number of hospitalizations solely attributable to this virus has decreased in comparison with the previous autumn-winter season.", but in Table 1 urgent codes are presented, not hospitalizations?

Author Response

The whole text should be checked by English native speaker before resubmitting it again.

Answer: Thank you. We further revise the English language of the manuscript.

The number of patients should be presented differently. I do not think, for example n.6, is appropriate way of presenting data.

Answer: Thank you for the correction. We revised the text accordingly.

It is stated "Table 2 summarizes clinical data according to virus group, comparing the impact of each virus on the urgent triage code and the need for respiratory support between the two years. In our cohort RSV is responsible for most of the urgent triage code during autumn-winter seasons 2021/2022 and 2022/2023, respectively n. 14 (38%) and n. 30 (49%)." However, when you analyze Table 2 it is clearly visible that in 31 (84%) cases during 2021/22 Rhinovirus was responsible for urgent triage code.

Answer: Thank you for your observation. Surely, Rhinovirus was involved in a significant percentage of critical infections. We highlighted this data in the results’ section.

I do not understand clearly difference between urgent hospitalization and urgent triage codes? It is stated "the number of hospitalizations solely attributable to this virus has decreased in comparison with the previous autumn-winter season.", but in Table 1 urgent codes are presented, not hospitalizations?

Answer: Thank you. Table 1 deals with the patients admitted to the ED and then hospitalized, being this our study population. We added more clinical data in Table 2 such as urgent triage codes. Urgent triage codes defined the severity of the clinical picture at the time of ED admission and prompt early medical evaluation.

This manuscript is a resubmission of an earlier submission. The following is a list of the peer review reports and author responses from that submission.

Round 1

Reviewer 1 Report

Comments and Suggestions for Authors

The paper describes a comparative analysis between the impact of RSV, Influenza virus and SARS-CoV-2 on urgent hospitalizations during autumn-winter 2021/22 and 2022/23. The manuscript is ineteresting for the purpose, but it is at a preliminary level, as a lot of significant information are missing.

For instance, the Authors state that no data on flu vaccination were available. Indeed this is significant to understand the evolution of influenza virus and also of RSV during SARSCoV-2 pandemic. No data about SARSCoV-2 vaccination are reported. How does SARSCoV-2 have influenced RSV and influenza epidemiology and vice versa?

No comment about the few cases of comorbity is reported. Have coinfections caused a greater severity of the disease?

Was the population considered affected only by a virus infection or also by other virus indipendent comorbidities? This aspect must be esplicited and discussed.

A more appropriate analysis of the data collected should be performed to catch the interest of the reader. No information about the genetic characteristic of the isolates are presented. 

Finally the Authors' own thought should be pointed out; the sentences are often presented in a too generic fashion.

Comments on the Quality of English Language

The English style is good and sound.

Reviewer 2 Report

Comments and Suggestions for Authors

The manuscript is presenting the differences regarding urgent hospitalizations related to Influenza virus, Respiratory Syncytial Virus and SARS-CoV-2 in children during two consequtive seasons. Everybody involved in clinical pediatrics was aware of this, however, observed things need to be reported and supported with exact numbers. Therefore, this article is valuable scientific addition to the existing knowledge in this field.

My minor remarks:

- in Introduction, besides ref. 6-8, I suggest adding ref. https://pubmed.ncbi.nlm.nih.gov/36560751/, as well.

- I suggest to change the last sentence of the Introduction section (The aim...) to make the time period more in concordance with the title of the manuscript.

- Rhinosiruses were most prevalente in both observed periods. However, their impact was not discussed in the manuscript. I suggest to add that.

- In Results, I am a little bit connfused with the numbers in Table 1. 27+3+22+38+3+8=101. 101>97. Are the coinfections reason for this discrepancy? Please, recheck everything and elaborate more clearly the distribution of the patients.

- Also, in Discussion, additionaly elaborate and add more data supporting the statement that "our data provide support for a change in clinical severity of RSV bronchiolitis..."

Reviewer 3 Report

Comments and Suggestions for Authors

The autumn/winter season is strongly connected with the flu-like infections, caused by many different viruses, as well as bacteria. Peri and colleagues present in their manuscript an epidemiological analysis of the respiratory pathogens in children, who required an urgent hospitalizations at two different time points. Unfortunately, there are no more data on the infectious diseases agents, such as sequencing, genotyping or neutralizing antibody titers. Thus, with great regret I need to say, that in my opinion the manuscript presenting only an epidemiological data does not suit to the Viruses. On the other hand, the presented data may be of interests for some readers, and thus I suggest to transfer the manuscript to other MDPI journals with lower Impact Factor, such as ‘Infectious Disease Reports’ or ‘Tropical Medicine and Infectious Disease’. Moreover, below you can find some additional comments, which may strengthen the manuscript.

MAJOR COMMENTS

1.       Authors focused mainly on three viruses, i.e. influenza viruses, RSV and SARS-CoV-2, but the highest number of children urgent hospitalizations  was caused by the rhinoviruses. Thus I recommend to change the title to: ‘Urgent hospitalizations related to viral respiratory diseases in children during autumn and winter seasone 2022/2023’. Moreover, authors should include some additional information on the rhinoviruses in the Introduction, as well as Discussion.

2.       What do the authors understand as social distancing measures? It needs to be clarified. Does this term include the face masks and common hand disinfection? What was the anti-COVID-19 politic in the Italy during autumn/winter seasons in 2022 and 2023? And how this could affect the prevalence of viral respiratory diseases.

3.       The number of urgent hospitalizations with positive NP swab, i.e. N=97 in season 2021/2022 and N=226 in season 2022/2023 are not equal with the data of particular infections, i.e. 27+3+22+38+3+8+5+1+2+1=110 in season 2021/2022 and 57+48+10+75+7+14+1+15+1+4+11=243 in season 2022/2023

4.       Finally, the English of the manuscript needs to be improved.

MINOR COMMENTS

a.       Please explain the abbreviations, such as HFNC in the Abstract.

b.       Please be consistent with the names of the infectious agents, i.e. influenza virus, SARS-CoV-2 and the names of the diseases, i.e. influenza, COVID-19.

c.       In the Table you present also the data on the prevalence of B. pertussis, which is not a viral agent

Comments on the Quality of English Language

The English must be improved.

Reviewer 4 Report

Comments and Suggestions for Authors

Manuscript: "Urgent hospitalizations related to Influenza virus, Respiratory Syncytial Virus and SARS-CoV-2… etc. ", by Francesca Peri et al.

The v2 manuscript is a well-meaning effort, and the authors appeared have made changes that are in red color. But there are fundamental issues that reduce this reviewer's enthusiasm about the manuscript, as summarized below:

1. The basic weakness is that the conclusions are already known!! The world knows that CoVID-19 is all but gone, and life has returned to the usual infections of seasonal flu, RSV etc. The Governments and pharmacies have spread out the SARS-CoV2 boosters farther apart, focusing more on the influenza vaccines. A new and effective RSV vaccine, the first one that appears to be more successful than any in the past, has just come out. So, there is really nothing new in this manuscript, The last sentence is in fact known to every citizen of this world: "Certainly, the SARS-CoV-2 pandemic impact will remain well imprinted on our mind, but pediatric data suggest that it’s time to focus again on well known diseases and new perspectives".

2. The paper is very haphazardly written. The logic often does not flow well. English is so poor that some sentences are hard to understand. It is also carelessly written, with typographical errors in nearly every sentence, such as the following in just one red area: "plausible to say", Sars-CoV-2 (needs to be capitalized), din't, didn't etc ("din't" is obviously wrong; "n't" is not proper in formal English, it should be "did not"), also note the "it's" in the last line, which should be "it is".
3. The authors acknowledged that the small number of the patients did not allow sub-classifications, but provided no solution. They are not willing to delay the manuscript to collect more data.

Overall, I do not see how the authors can even improve the paper, except perhaps the writing and the English. But that will only improve its readability, not its scientific value.

Minor: Replace the abbreviation HFNC in the Abstract with the full name, since it is a very specialized clinical term, not commonly used. The References are also carelessly formatted. Next time, please add line numbers, so that the reviewers can describe the exact location of the errors.

Comments on the Quality of English Language

See the detailed comments to the Authors.

Round 2

Reviewer 1 Report

Comments and Suggestions for Authors

The Authors added some comments, but overall the paper is very generic and thus less informative.